# Relationship Dynamics among Couples Dealing with Breast Cancer: A Systematic Review

**DOI:** 10.3390/ijerph18147288

**Published:** 2021-07-07

**Authors:** Marco Valente, Ilaria Chirico, Giovanni Ottoboni, Rabih Chattat

**Affiliations:** Department of Psychology, University of Bologna, 40126 Bologna, Italy; marco.valente4@unibo.it (M.V.); ilaria.chirico2@unibo.it (I.C.); giovanni.ottoboni@unibo.it (G.O.)

**Keywords:** breast cancer, dyads, couple, psychological adjustment

## Abstract

Most studies have been concerned with the experiences and needs of women with breast cancer and spouses/partners separately. In this review, the relationship dynamics that characterize the couple’s experience of breast cancer treatment were investigated. Findings will inform both researchers and professionals in the area of oncology. A systematic literature search was performed in CINAHL, PsychINFO, MEDLINE, Scopus and Web of Science. A checklist for qualitative and observational studies was used to evaluate the methodological quality of the studies. Seventeen studies were included, and the synthesis of the literature revealed five domains that characterized the dyadic process: dyadic coping strategies, psychosocial support, communication, the couple’s sexual life and spirituality. The included studies provide the basis for knowledge and awareness about the experience of couples with cancer, the specific dimensions enacted during the breast cancer treatment path and the type of responses that are associated with a positive couple’s adjustment to the disease.

## 1. Introduction

In women, breast cancer is the most common disease in incidence (2.1 million new cases in 2018) and mortality (627,000 deaths in 2018) when compared to all cancers [1,2].

In the past century, most women did not survive breast cancer [3], but, more recently, earlier diagnosis, effective screening programs and advanced medical treatments have been increasing the number of survivals in a 5-year rate [4].

Traditionally, in the oncology literature, greater attention has been paid to the psychiatric, psychological and psychosocial impact of the pathology. Several studies analyzed the adaptive reaction strategies of patients at the medically defined points such as diagnosis disclosure, treatment consents, disease recurrence and palliative cares initiation [5,6,7].

According to existing literature, the issue of breast cancer survivorship brings attention to the end of treatment perspective [8,9,10]. In this regard, Carter [11] analyzed longitudinally the daily lives experience of 25 women aged from 40 to 78 years who had survived breast cancer. Participants highlighted how the end of the treatment represents a crucial moment during which they can retrace the cancer pathway: interpreting and understanding the diagnosis, confronting the idea of mortality, rearranging their life priorities, coming to terms with the diagnosis, being able to move on and flashing back to the experience. Thus, according to Carter, ‘going through’ this pathway suggests a past, present, and future life after the diagnosis for those affected.

From the early nineties, the awareness about the involvement of patients’ partners in the experience of cancer has become more evident [12,13]. Hence, a considerable body of studies has focused on the breast cancer psychological consequences for partners and other family members [14,15,16].

This has gradually led to consider breast cancer as a ‘*we-disease*’ [14], which takes shape in the context of the relationship [17]. In short, literature reviews and meta-analysis support the idea that dyads may react as a unit rather than as individuals when coping with cancer, which influences the distress experienced by both partners [18,19] and might impact their relationship functioning.

In this context, the dyadic approach has progressively entered the research studies’ design [20], and several studies in the breast cancer literature have investigated the correlations between a specific couple dimension and psychological adjustment [21,22,23].

In this context, just a few reviews have focused on the impact of breast cancer on the couple’s relationship [15,18,24,25]. They mainly reported the areas of interaction of breast cancer and the patient’s personal relationships: (1) general interactional themes of patients secondary to breast cancer (e.g., social isolation; victimization; uncertainty); (2) the impact of social support, including the support from partners on patients’ psychological adjustment; (3) the impact of the disease on the parent–child relationship. Moreover, Staff [25] highlighted the dimension of dyadic coping by exploring the antecedents and the outcomes in close personal relationships. In addition, these reviews discussed the impact of breast cancer on marital level of satisfaction in terms of couples’ sexual relations and communication. Despite the emerging evidence on the impact of breast cancer on patients and partners, no systematic review has been conducted specifically on the relationship dynamics that can have a major impact on the psychological distress experienced by couples. Consequently, the first aim of this review is to highlight and summarize the relationship dynamics that characterize couples during the breast cancer treatment pathway after the acute phase of treatments. Furthermore, we want to clarify which areas might predict a positive trajectory of psychological dyadic adjustment.

We used interdependence theory [26] and the investment model of commitment [27] to guide the development of the current study. Developed to clarify behaviors in dyadic relationships, the interdependence theory assumes that partners become interdependent over time through their interactions. The investment model implies that interdependence will be perceived as commitment, defined by partners’ desire to maintain the relationship through good and bad times [27].

## 2. Materials and Methods

This systematic review was performed in accordance with the Preferred Reporting Items for Systematic Reviews and Meta-analyses guidelines (PRISMA) [28].

### 2.1. Inclusion Criteria

To be eligible, the articles had to: (a) be focused on the dyads consisting of women and their partners/spouses; (b) deal with women who had finished the course of breast cancer treatment (e.g., radiotherapy, chemotherapy or surgery) for at least 6 months; (c) address the process of dyadic adaptation or adjustment after the treatments. We excluded articles that specifically dealt with either support or intervention evaluation, that studied causes of cancer, diagnostic or other medical considerations, or that discussed disease prevalence or incidence. Moreover, we excluded studies that did not present data separately for breast cancer patients. Additionally, only articles written in English and published in peer-reviewed journals were included. Theses, dissertations, conference proceedings, or trial registries were not considered.

### 2.2. Search Strategy

The literature selection, up to February 2020, involved the following electronic databases: CINAHL, Pychinfo, MEDLINE (via Pubmed), Scopus and Web of Science. We decided not to restrict the publication dates to increase the number of potential studies identifiable. The search strategy (Table 1), a combination of keywords and MeSH terms blended was “*breast neoplasm*” or “*breast cancer*” or “*breast carcinoma*” AND “*couples*” or “*dyads*” or “*women and their partners*” or “*spouses*” AND “*relationship quality*” or “*changes in relationship*” or “*social, adjustment*” or “*adaptation, psychological*”. To complete the search strategy, we used the Thesaurus to identify variations in search terms. Finally, we screened the reference lists of the included studies to check for additional eligible studies.

### 2.3. Study Screening

The Mendeley software was adopted to manage all the articles (both included and excluded ones) analyzed in the research process.

After checking study duplications, two reviewers (I.C. and M.V.) analyzed all articles based on titles, keywords and abstracts according to the inclusion criteria to select proper studies. Any discrepancy was resolved by a third reviewer (G.O.) through discussion until an agreement was reached. In the selection process, any articles for which it was not clear whether they should have been included were moved to the next step. Finally, the full-text articles that likely matched the inclusion criteria were reviewed independently by two members of the review team (I.C. and M.V.) with reasons for exclusion annotated; again, any discrepancy was resolved by a third reviewer (G.O.).

### 2.4. Data Extraction

The data extraction form followed the Cochrane Collaboration model [29]. The compiled form identified key issues of each study such as the authors, year of publication, country, number of dyads, design of the study, measurements points, investigated variables and key results. By using these key issues, all study outcomes were clustered, resulting in several themes to describe the results.

All the included studies were equally distributed between the authors (M.V., I.C.), who independently extracted the data from the articles. Any disagreements were resolved by consulting the other authors (G.O., R.C.) to make the final decision.

### 2.5. Quality Assessment

The methodological quality of all studies was independently assessed by two reviewers (M.V., I.C.). Any disagreements were resolved by a third reviewer (G.O.), and a full agreement was reached.

To evaluate the quality of the quantitative studies, we used the quality checklist with 23 criteria [30], while for qualitative studies, a quality checklist with 12 criteria [31] was used. If the criterion was met, it was rated with a “+” and unmet with a “−”, and when the criterion was not completely met, it was rated with “+/−”. If the criterion was not applicable, it did not receive a rating.

## 3. Results

Figure 1 reports, through a flow chart, the selection process used to identify articles for inclusion. The search in the five electronic databases resulted in 2094 articles after removing 925 duplicates. After revising titles, keywords and abstracts, we excluded further 2018 articles based on the title or abstract. We then assessed the full texts of the remaining 76 articles. Due to the established inclusion criteria, 60 articles were excluded for the following reasons: they only focused on a specific dyadic dimension such as dyadic coping, psychosocial support, general interactional themes; they dealt with the psychological functioning of patients or partners; they included other family members than partners without separate data analyses; they investigated the outcomes of a dyadic intervention. Full texts of four studies were not available. After the reference lists were screened, one additional article was found by means of cross-referencing; thus, 17 articles were finally included in the present review.

### 3.1. Quality Check

Table 2 and Table 3 reported the quality ratings of the included studies.

For the qualitative studies (Table 2), the total score on the quality checklist ranged from 7.5 to 10.5 out of 12. Most studies received a high score, while those with lower quality reported a limited description of the sample and/or the sampling method. Specifically, not every study explicitly described participant characteristics. Description of the sampling and justification for the sampling strategy was not always reported, thus making it difficult to evaluate the quality of the sample. Moreover, relevant aspects associated with the research process, such as the information about researcher reflexivity, the researcher’s potential influence on the research process and if/how related problems were dealt with, were not addressed in some studies. Similarly, the researcher’s ongoing commentary and critical reflection of study biases and assumptions and how these have influenced all stages of the research process were omitted or not fully provided.

For the quantitative studies (Table 3), the total score on the quality checklist ranged from 15 to 17.5 out of 23. The lowest scores were mainly due to small sample sizes, participants were not always representative of the population, and inclusion and exclusion criteria were not fully clear. Moreover, there was a lack of description of the procedures aimed to control for possible confounders.

### 3.2. Study Characteristics

The general characteristics (authors, year of publication, country, number of dyads, design of study, measurements points, investigated variables and key results) of the 17 included articles is presented in Table 4. The studies were published between 1989 and 2019. Most studies were conducted in Western countries: USA (*n* = 10), France (*n* = 2); Canada (*n* = 1), Turkey (*n* = 1) and Australia (*n* = 1). Only one study took place in the Republic of Korea. The reviewed studies included populations belonging to different ethnic groups. The majority of studies concerned the Caucasian population, and one study included African American couples. Another study placed the Asian population at the center of its research design.

For what concerns population features, sample sizes were remarkably wide, spanning from 7 to 282 participants. They were middle-aged (range: 45–55 years old). All studies were conducted with heterosexual couples. As regards the medical condition, most of the women enrolled in the studies had a diagnosis of malignant breast cancer. One study included women with both malignant and benign breast cancer. The range of months since the end of treatment was variable as it spanned from 12 to 60 months. In only one study, women were currently undergoing hormone therapy.

As regards the study design and study measures, adequate study designs were chosen in relation to the objectives of the studies. In total, nine papers used a qualitative approach, seven papers used an observational quantitative approach, and one paper used both quantitative and qualitative measures design. Moreover, the questionnaires were psychometrically accurate, the qualitative techniques seemed to be adequate, or appropriate statistics were used. Nevertheless, most studies included an appropriate number of couples to answer their research questions.

### 3.3. Findings

#### 3.3.1. Relationship Dynamics

Different areas of relationship characterize the dyadic process of psychological adaptation to breast cancer. The deployment of shared coping strategies, defined as *joining forces* to face the problem, and the availability of instrumental and emotional support have been frequently mentioned in the studies [32,33,34,35,36,37,38]. In regard to coping strategies, couples who adopt common dyadic coping strategies, consisting of the attempt of one member of the dyad to reduce the stress perceived by the other member and as a common effort to cope with the situation, report less psychological distress [38] and high levels of adaptation to the pathology [32]. In this context, a crucial role is played by the merging strengths [33] that are defined as the capability of the couple to join efforts and collaborate to face the challenges of the diagnosis (e.g., walking together, searching for information and support networks, trusting together, staying together).

The dimension of psychosocial support, both instrumental and emotional, is regarded as fundamental in the process of a couple’s adaptation to breast cancer. Despite its relevance, dyads facing malignant breast cancer often report a decline in psychosocial support over time [39]. In most cases, requests for support were the result of a mutual process of understanding each dyad’s member’s own personal needs. When available, the presence of an informal and formal social network is helpful to provide support from different points of view, thus avoiding experiences of isolation and helplessness towards the disease. Skerret [32] highlights how the nature of psychosocial support is also characterized by multigenerational legacies. In this regard, collecting information about the personal history of each member of the dyad might work to fashion personal theories about a coping modality. This aspect is found as a positive prognostic factor for the couple’s adaptation to the disease. Furthermore, psychosocial support is reported as a fundamental dimension to assist the couple in maintaining a positive dyadic relationship, both in the acute phase of the treatment and during the early survivorship phase [37].

Another aspect suggesting a positive adaptation to the disease and is strongly related to the functioning of relationships is represented by communication. A positive and complementary pattern of interactions favors the dyad in facing the breast cancer care pathway, and this, in turn, enhances the likelihood of positive emotions [11,32,33,35,37,38,39,40]. Moreover, those couples reporting a conflict resolution through negotiations have lower levels of discomfort. Additionally, greater communication regarding stressors due to the illness increases the levels of relationship satisfaction and potentially modulates the perception of the patient’s physical damage [41]. For what concerns the trajectory of psychological adjustment, communications based on navigating potentially hurtful disclosures and responding to partners’ obstructive behaviors seem to favor the couple’s adaptive efforts [42]. On the contrary, couples who report greater avoidance of discussing problems and stressors or greater use of request-withdrawal communication experience higher levels of suffering and psychological impairment [41].

Since treatments have tangible consequences on women’s bodies, those aspects also enter the couple’s experiences of uncertainty and worries about the impact of the disease on the relationship [34,42,43]. Moreover, changes in women’s physical appearance shape the couple’s communication. Those communications sustaining and strengthening survivors’ body esteem seem to predict a greater relational satisfaction [42].

Closely related to a woman’s personal transformations is the couple’s sexual life. This aspect appears to be influenced by ethical concerns about the dissatisfaction with sexual relationships and fears of sexual health treatments [42]. Moreover, high resilience in sexuality is predictive of a positive couple’s adaptation to the disease [32].

Religiosity is another dimension relevant in the path of treatment for breast cancer that influences the marriage union [35,43]. Not only sharing similar religious beliefs (faith and belief in the power of God) but also the possibility of praying together are predictive factors for a better relational functioning during the trajectory of the illness, particularly in the African American population [35].

#### 3.3.2. Diagnosis and Acute Phase of Treatment

The first couple’s experience with breast cancer is represented by the diagnostic stage. The etiopathology of cancer has contributed to making this disease mysterious, unknown and together with high incidence rates, a halo of fear has gradually emerged around the word tumor. The gloomy and deadly image accompanying this pathology leads couples to react to this diagnosis with feelings of uncertainty, shock and exacerbating distress [43].

Responses given by couples facing either benign or malignant diseases are clearly dissimilar. Couples in treatment for malignant cancer refer more uncertainty about the nature and course of the disease than couples facing benign ones [39]. Couples in the malignant group report significantly higher levels of emotional distress and more psychosocial problems (e.g., role function, responsibilities, activities of daily living) than couples in the benign group, and these differences persist over time [39]. Furthermore, for a successful adaptation to the disease, couples need to clearly understand the amount and quality of available resources since the stage of diagnosis [32].

The emergence of either initial side effects or post-operative difficulties can have pathological consequences on couples. For instance, women’s symptoms could have an impact, not only a physical one but also an emotional and psychological one [34,35,36]. Furthermore, acute treatments could potentially exacerbate the couple’s feelings of the effectiveness of the treatments. The two factors associated with the acute phase of treatment that seems to contribute to disease management are the articulated beliefs on health and disease and those related to the ability to control or influence the course of the disease [32]. On the contrary, the psychosocial difficulties could cause the couple a sense of misunderstanding each other’s worth and needs [43].

Hence, going through the breast cancer pathway seems to strengthen the couple relationships in certain conditions [44]. On the other hand, couples seem to exhibit anxiety-related feelings towards their future. Consequently, emotional issues about cancer recurrence may appear, and the difficulty of dealing with such long-term perspectives is exhibited. Moreover, family planning, changes in roles and health-promoting behaviors will remain present despite the positive treatment outcomes. Marital satisfaction and family functioning seem to be predicted by four factors: advice on how to cope with breast cancer by women’s partners; their physical and emotional presence for women undergoing surgery; more tender and affectionate behaviors towards women after diagnosis; and partners’ perceptions of patients as confidant [44]. In this regard, Carter’s [11] highlighted those marriages that are primarily characterized by extraordinarily high levels of involvement between the two couple members and less rigidity in terms of interaction based on role flexibility.

#### 3.3.3. Impact on Each Member of Dyad

The impact of breast cancer seems to involve both partners in socio-demographic, medical and psychological aspects.

Among the most studied individual aspects, psychological distress seems to be the strongest predictor of the quality of life, a variable capable of predicting couple’s adaptation skills, which are strongly related to mental health [45,46]. Northouse [45] analyzed the correlations between socio-demographic and medical factors, psychosocial and mood symptoms and psychological distress. At 18 months after the breast surgery, approximately 35% of patients and 24% of partners reported moderate levels of distress. Both of them had few difficulties with carrying out duties at work, family and social circumstances. For what concerns the sample of partners, the demographic factors were related to mood disturbances. Specifically, younger men and in marriage for a shorter period of time reported fewer positive states. Additionally, partners of women with recurrent cancer or women on chemotherapy report more role marriage-associated problems.

In Kim’s study [46], patients and partners mostly showed similar levels of psychological distress and quality of life. Moreover, a partners’ levels of distress and the dissimilarity between couple members regarding psychological distress were found to significantly influence each one’s quality of life [43].

Carter [11] considered some predictive variables of the adaptation of the individual members of a couple, evaluating if and how they were prognostic of a dyadic adaptation. No apparent psychological consequences took place over time to either the women or their partners as a single individual, and no significant differences emerged between them on the individual adjustment to illness.

In terms of coping strategies, for both husbands and wives, a greater relative use of seeking support and less use of avoidance is associated with better psychological well-being [47]. Lower levels of well-being are also related to the women’s use of problem-focused coping and wishful thinking and a partners’ tendency to blame themselves [47]. Furthermore, a consistent pattern of correlations among coping techniques for members of the dyad occurs, although the self-reported coping of one member is largely unrelated to the self-reported coping of the other one.

Some studies underline the women’s personal experience in relation to the pathology, “*what the disease revealed to me*” [36], and how this affects couple relationships once the acute phase of treatment has ended. Notably, the attitude of survivors to prioritize their own needs has an impact on women’s partners and the relationship functioning [37]. What happens is that this experience represents a turning point for women to change their attitude from being introspective or forbearing to being more assertive and expressive [35].

## 4. Discussion

This systematic review shows the dynamics that characterize couples during the breast cancer treatment pathway by underlining the areas that can predict a positive dyadic adaptation. More specifically, couples who maintain a resilient intimacy, build constructive mutual communication, adopt common dyadic coping strategies and provide mutual psychosocial support report higher levels of relationship functioning. These marriages are characterized by high levels of enmeshment and less rigidity in terms of rules and roles and flexible interaction patterns.

According to the breast cancer literature [48,49], diagnosis disclosure is experienced as a shock by couples mostly because of its terminal nature [35,43]. Breast cancer treatments, including surgery, chemotherapy, radiotherapy and hormone therapies, have strong effects on women’s bodies and their images and directly influence sexual functions and hormone levels [50]. Often women with breast cancer need endocrine adjuvant treatment that involves insufficient lubrication, dyspareunia and sexual arousal [51]. Along with the difficulties about sexuality, the consequences of a compromised dyadic intimacy add to a decrease in mutual support and marriage satisfaction [52]. Resilient intimacy, as a unique dyadic process, is now considered a predictive factor for a couple’s positive psychological adaptation to cancer and other health adversities [32,53].

Communication within the couple can undoubtedly impact how the dyad copes with treatments [54]. Furthermore, couples who can develop constructive mutual communication, explore potentially hurtful disclosures, respond to partners’ obstructive behaviors regarding cancer-related issues seem to have a more positive psychological adjustment to the breast cancer experience [32,40,42]. The relational dimension of communication seems to be closely connected with different outcomes. First, it is important to highlight that open and shared communication about potential fears, worries or needs allows couples to share the experience with higher levels of satisfaction [17] and more successful coping efforts [21]. Furthermore, with reference to the optimal matching model of social support, adopting a complementary communication consisting of a mutual sharing of the emotional aspects of the care pathway increase the possibility of better matching of needs and reciprocal support [55].

In the oncological literature, the dimension of dyadic coping is defined as “*the interplay between the stress signals of one partner and the coping reactions of the other, a genuine act of shared coping*” [56] is considered as highly relevant for relationship outcomes in couples dealing with cancer. Couples deal with many potentially stressful challenges, such as emotional concerns and existential issues, medical treatment and its side effects, transformed sexuality and changed social relationships and roles during and after treatment. Longitudinal studies have shown how dyadic coping is identified as a protective factor for the couple’s relationship [56] and a predictor of a more positive couple’s psychological adaptation by protecting the dyad’s quality of life [25]. Moreover, the relational factors can impact the couple’s coping behaviors and, ultimately, the success of their coping efforts [19]. This appears to be in line with the results of this review showing how the couple’s ability to merge forces plays an important role in the dyadic coping strategies [35].

As pointed out in several studies [49,57,58], psychosocial support plays a key role in the treatment pathway for breast cancer. Beyond a supporting couple relationship, at the emotional and instrumental levels, the informal network plays a crucial role as well. Psychosocial support has a substantial influence on the adaptation process to cancer diseases [59]. Receiving psychosocial support from social networks increases self-esteem, reduces the stress associated with the disease and improves adherence to medical treatments [57]. In particular, the increase in treatment adherence can be determined by improved cognitive functioning, a sense of self-efficacy, intrinsic motivation, personal control, reduced emotional conflicts, distress and depressed mood [60]. However, the stress-buffering model argues that psychosocial support enhances well-being when the individual is highly distressed, thus protecting people from the pathogenic effects associated with psychological distress [61]. According to this model, psychosocial support can act in two ways. First, it intervenes between the stressful event and the experience of psychopathological outcomes while preventing and mitigating stress responses. Secondly, it can intervene between the stressful experience and quality of life outcomes by directly influencing the health-related behavior.

In addition to communication, coping strategies, intimacy and psychosocial support, religiosity has been identified as a relevant dimension in couples’ dealing with breast cancer. Evidence shows that religiosity can have a positive impact on patients’ lives consisting of a decrease in negative emotional states, levels of distress, mood symptoms and hopelessness and an improvement in well-being and illness adjustment by promoting reflection and reconceptualization of the situation [14,33,62]. Furthermore, in regards to the African American population, praying together and, more generally, taking care of their own spirituality seems to favor couples’ management of the stressful situation [63].

Research focusing on dyads as a unit of analysis represents a quite recent field of oncology, with most studies conducted in the twenty-first century, while previous research has dealt with the patient and the partner as independent entities [64]. Personality traits can be considered additional factors implicated in the relational functioning of couples facing the treatment pathway for breast cancer. The National Comprehensive Cancer Network (NCCN) has identified distress as “*a multifactorial unpleasant emotional experience of a psychological, cognitive, behavioral, emotional, social, and/or spiritual nature that may interfere with the ability to cope effectively with cancer, its physical symptoms, and its treatment*” [64,65]. High levels of distress seem to negatively correlate to all-cause and cancer-related morbidity and mortality, as well as to the quality of life [65]. Some studies included in this review [45,46] have focused on the associations between socio-demographic, medical, psychosocial factors and mood symptoms, levels of distress and quality of life of each member of the dyad. Results have shown that high levels of distress and impaired quality of life are negative prognostic factors on the couple’s ability to adapt. Coping strategies adopted by the two individuals, as two separate entities, and their concordance potentially impact on couple’s adjustment [46]. In the case of breast cancer, partners’ coping strategies focused on avoidance and self-blaming appear to have negative effects on the couple’s well-being. Similarly, the women who adopt wishful thinking as a coping strategy show higher levels of distress. Literature in this field highlights how the use of coping strategies by the members of the couple, which are exclusively centered on the problem, seems to favor higher levels of well-being and better psychological adaptation [66,67].

## 5. Conclusions

According to existing literature, a high relationship functioning during or after cancer treatment may depend on how properly the dyad incorporates and psychologically elaborates cancer issues into their lives. This highlights the need to increase awareness and consider breast cancer as a dyadic affair with a significant impact on the relationship. Findings from this systematic review shed light on the significant impact of breast cancer on relational functioning, showing how breast cancer impacts relational dimensions and on the complex interplay between partners.

Moreover, facing cancer as a ‘we-disease’ may result in a strengthening of the couple’s relationship. Indeed, considering breast cancer as a relational disease can facilitate the implementation of successful dyadic interventions aimed to develop or consolidate the relationship functioning. The positive implications of such an approach may be different: a greater awareness about the impact of the disease, the maintenance of a positive relationship functioning, the improvement of the compliance with care pathways, higher quality of life and better coping strategies in facing with disease fostering the process of psychological adaptation.

To our knowledge, this is the first systematic review that brings together and analyzes almost 30 years of studies in this field, thus summarizing the relationship dynamics that characterize couples during the breast cancer pathway after the acute phase of treatments. Although the review was rigorous and wide-reaching, there were a few limitations. We only searched for publications mentioned in English in peer-reviewed journals, theses, dissertations, conference proceedings and trial registries were excluded. Moreover, the study samples were small and not always fully described, and most studies had a cross-sectional design.

In conclusion, breast cancer strongly impacts the entire dyadic system. Clinicians and other health professionals can play a vital role in helping the couple to adjust to the psychological and psychosocial effects of breast cancer by adopting a collaborative approach and including dyads in the clinical consultations. Further research in this area should encourage new theoretical frameworks for the development of specific couple interventions to promote a positive psychological adjustment to the disease and, consequentially, to maintain the health of these relationships.

## Figures and Tables

**Figure 1 ijerph-18-07288-f001:**
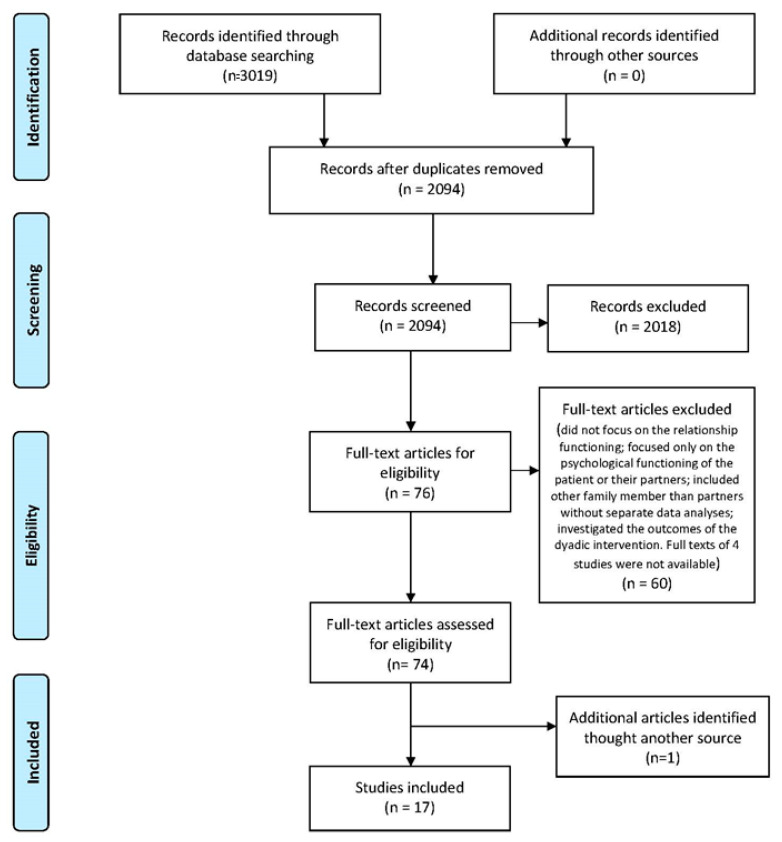
PRISMA flow diagram of study inclusion in the review.

**Table 1 ijerph-18-07288-t001:** Search strategy.

Search Terms
Breast neoplasm OR Breast cancer OR Breast carcinoma (title or abstract) Couples OR Dyads OR Women and their partners OR Spouses (title or abstract) Relationship quality OR Changes in relationship OR Social, adjustment OR Adaptation, psychological (title or abstract) #1 AND #2 AND #3

**Table 2 ijerph-18-07288-t002:** Quality assessment of the included quantitative studies.

	Northouse, 1989	Carter, 1993	Ptacek et al., 1994	Hoskins, 1995	Northouse et al., 1998	Meier et al., 2019	Kim et al., 2008	Manne et al., 2006
Accurate and appropriate outcome (intervention) measures in all participants	+	+	+	+	+	+	+	+
Adjustment for confounding	+	+	+	+	+	+	+	+
Case/controls recruited from same population	−	−	−	−	+	−	−	−
Appropriate statistical tests used	+	+	+	+	+	+	+	+
Participants representative of population	+	+	+	+	+	+	+	+
Potential confounders described	−	−	−	−	−	−	−	−
Recruitment of case/control over same time frame	−	−	−	−	−	−	−	−
Participants characteristics described	+	+	+	+	+	+	+	+
Numerical description of important outcomes given	+	+	+	+	+	+	+	+
Outcomes clearly described	+	+	+	+	+	+	+	+
Response/nonresponse rate described	+	+	+	+	+	+	+	+
Clear case/control definition	−	−	−	−	+	+	−	−
Power calculation used	N/A	N/A	N/A	N/A	N/A	−	N/A	N/A
Losses and completers described	+	N/A	+/−	+	+/−	−	−	+
Reliable assessment of disease state	+	+	+	+	+	+	+	+
Clear inclusion/exclusion criteria	+	+	+	+	+	+	+	+
Clear hypothesis	+	+	+	+	+	+	+	+
Reported probability characteristics	+	+	+	+	+	+	+	+
Type of study stated	+	+	+	+	+	+	+	+
Main findings described	+	+	+	+	+	+	+	+
Disclosure of funding source	N/A	N/A	N/A	N/A	N/A	+	N/A	N/A
Conclusions supported by findings	+	+	+	+	+	+	+	+
Statistical tests of heterogeneity	−	−	−	−	−	−	−	−
Total Score	16	15	15.5	16	17.5	17	15	16

**Table 3 ijerph-18-07288-t003:** Quality assessment of the included qualitative studies.

	Antoine et al., 2013	Skerrett, 1998	Picard et al., 2005	Keesing et al., 2016	Chung et al., 2012	Canzona et al.,2019	Dorval et al., 2005	Morgan et al., 2005	Cömez et al., 2016
Clear statement of, and rationale for, research question/aims/purposes	+	+	+	+	+	+	+	+	+
Study thoroughly contextualized by existing literature	+/−	+	+	+	+	+	+	+	+
Method/design apparent, and consistent with research intent	+	+	+	+	+	+/−	+	+	+/−
Data collection strategy apparent and appropriate	+	+	+/−	+/−	+	−	+	+	+
Sample and sampling method appropriate	+/−	+	−	+	+	+	+	+	+
Analyticapproach appropriate	−	+	−	+	+	+	+/−	+	+
Context described and taken account of in interpretation	+	+	+	+	+	+	+	+	+
Clear audit trailgiven	+	+/−	+	+	+	+	+	+/−	+
Data used to support interpretation	+	+	+	+	+	+	+	+	+
Researcherreflexivitydemonstrated	−	−	−	+	+/−	−	−	−	−
Demonstration of sensitivity to ethical concerns	N/A	N/A	N/A	N/A	N/A	N/A	N/A	N/A	N/A
Relevance and transferabilityevident	+	+	+	+	+	+	+	+	+
Total Score	8	8.5	7.5	10.5	10.5	8.5	9.5	9.5	8.5

**Table 4 ijerph-18-07288-t004:** Characteristics of included studies.

Author	Country	N of Dyads	Design	Measurement Point/Intervals	Variables	Key Results
Northouse, 1989	USA	41	Observational	At three points in time: 3 days, 30 days, and 18 months post-surgery	Mood, roles functioning and symptoms over distress as post indicators of adjustment.	At 18 months post-surgery, approximately 35% of patients and 24% of partners reported a moderate level of distress. Both patients and partners had few difficulties carrying out the role functions at work, family and social levels. No correlation between demographic factors (e.g., age, level of education, length of marriage) and adjustment was found. Furthermore, no significant differences resulted between medical factors (type of surgery, evidence of recurrence, current treatment, completion of breast reconstruction) and psychological distress or mood symptoms. As for husbands, demographic factors were related to mood disturbance. Younger men and in marriage for a shorter period of time reported a lower level of positive states. Husbands of women with recurrent cancer or women on chemotherapy reported more role problems.
Carter, 1993	USA	14	Mixed Methodology	2–3 years from the mastectomy	How the couples adapt over the term of treatment. What interactional factors characterize couples affected by BC.	No apparent psychological consequences took place over time to either the husband or the patient as a single individual. Differences did not occur between husbands and wives on individual adjustment to illness. In contrast, significant differences were found in marital interactions that define the psychology of the couple system. Marriages are characterized by extraordinarily high levels of enmeshment. Less rigidity, in terms of rules and roles, flexible interaction patterns (three components: marital domination, the punitiveness of one spouse toward the other and conflict resolution through negotiations), communication patterns that facilitate disclosure and free exchange represent crucial components of adaptability.
Antoine et al., 2013	FR	11	Qualitative	Undergoing hormone therapy	Identify the experiences of partners and young women who had breast cancer, identify the marriage functioning in BC hormone treatment.	Five main themes emerged: the disease cemented our relationship; the mirror breaks (i.e., refers to physical, psychological and emotional symptoms); the onslaught of solidarity (means the support, both instrumental and emotional one, by close and distant family and friends); a suspended future (cancer and recurrence, difficulties with long term perspectives and family planning); what the disease revealed to me.
Skerrett, 1998	USA	20	Qualitative	From 18 to 31 months post diagnosis	How a diagnosis of breast cancer affects the marital relationship. How different aspects of the relationship can help or impede couple adaptation.	High Adapters: challenging impact identifiable, united coping philosophy, selective communication patterns, positive use of multigenerational legacies (i.e., the information about their history to fashion personal theories about a method of coping that would work for them), articulated beliefs on health and illness, and beliefs regarding one’s ability to control or influence the course of illness and well-being, resilient sexuality. At-Risk Adapters: devastating impact, lack of dyadic coping, mutual isolation, strained communication, difficulty using multigenerational legacies, absent or conflictual beliefs, multiple stressors (e.g., medical complications, past losses and traumatic histories).
Ptacek et al., 1994	USA	36	Observational	Currently disease free (1.5 years from the end of radiotherapy)	The couple’s modalities of coping with treatment.	A consistent pattern of correlations among coping techniques for both spouses was reported. Respectively, it was found a strong correlation, for both husbands and wives, among the coping strategies of self-blame, wishful thinking and avoidance. However, the self-reported coping of one spouse was largely unrelated to the self-reported coping of the other spouse. In terms of well-being, for both husbands and wives, the greater relative use of seeking support and less use of avoidance was associated with better mental health. A low level of well-being was also related to the use of problem-focused coping and wishful thinking in wives and blaming oneself in husbands. Satisfaction with the marital relationship was far less strongly correlated to cancer-specific coping.
Picard et al., 2005	FR	16	Qualitative	From 10 to 12 following the initial diagnosis	Ways in which the couple as a dyad deal with the disease and associated treatments.	Four themes: dealing with the Unknown; dealing with the Threat of Loss and the Uncertainty of the Future; dealing with the Woman’s Personal Transformations in the Couple’s Sex Life; organizing a Social Support Network.
Keesing et al., 2016	AUS	8	Qualitative	Till 6 months to 5 years from completing the treatment, excluding adjuvant hormone treatment	How wives and spouses communicate with each other; the pattern used by the couples to maintain their relationship; the needs and supports required by women and their partners.	Three themes: A disconnection within the relationship (the woman survivor of breast cancer needing to prioritize her own needs, sometimes at the expense of her partner and the relationship); Reformulating the relationship (i.e., reflects the strategies used by couples to negotiate changes within the relationship); Support is needed to negotiate the future of the relationship (i.e., couples emphasized the need for additional support and resources to assist them in maintaining their relationship during early survivorship).
Chung et al., 2012	KR	7	Qualitative	From 5 to 63 months following the initial diagnosis	The aspects of couples’ going throw after BC diagnosis.	Nine themes: HITTING A WALL (when the women and their husbands initially heard about the cancer diagnosis, a feeling of shock was the most common immediate response); FACING HARDSHIPS WITH TREATMENT AND SUPPORT (after their immediate responses, couples underwent a difficult phase of facing hardships with treatment and support as summarized with “suffering with treatment” and “feeling; CONTROLLING AND PROTECTING MYSELF (women focused on themselves: controlling and protecting myself); REFORMING MY LIFE TO CARE FOR HER (husbands try to find ways to help their wives more actively, three categories of “reforming my life,” “providing care” and “keeping a positive attitude”); WORKING TO SURVIVE THE REALITY (couples agreed to deal with the situation together as categorized by “following standards,” “accepting the new reality” and “working to survive”); COMING INTO MY OWN (cancer as a turning point for women to change their attitude from being introspective or forbearing to being more assertive and expressive, BEING A CARING PERSON (changes in husbands); KNOWING THINGS’ WORTH (the couples learned lessons, involving four categories of “appreciating partners,” “thinking what this event means,” “thinking about what is important” and “asking for further support”); BEING SUSPENDED WITHOUT RESOLUTION (several issues were still challenging the couple)
Canzona et al., 2019	USA	53	Qualitative	End the treatment from 3 months to 25 years prior to recruitment	How couples attempt and experience the challenging of BC.	Five sources of uncertainty: perceptions of post treatment bodies, worry about effects on relational partners, ethical concerns about dissatisfaction with sexual relationship (partner prospective), fears about future of the relationship and apprehension about Sexual Health treatment uselessness. Four themes of communication efforts: supporting survivors’ body esteem (partner prospective), navigating potentially hurtful disclosures, responding to partners’ obstructive behavior and believing communications useless.
Dorval et al., 2005	CA	282	Qualitative	Three measurement points: at 2 weeks and 3 and 12 months after treatment start	The potential adaptation Predictors from the perspective of both couple Members.	A global agreement between the patients and their spouses about the effect of the disease on their relationships emerged. Most of the couples reported that breast cancer and associated treatments had made them closer. In terms of marital satisfaction, at twelve months, significantly higher levels were found among couples where both partners reported individually that breast cancer had made them closer. Four factor predictors were found: giving advice to the spouse about coping with breast cancer, the spouse’s accompanying the patient to surgery, and the spouse reporting the patient as a confidant. The fourth factor consisted of tenderness and affection from her spouse since diagnosis.
Morgan et al., 2005	USA	12	Qualitative	Not specified field	The pattern by which African American couples cope with BC.	Two main dimensions: merging strengths (as uniting and working together to cope with the challenges of a breast cancer diagnosis) to cope with and survive a breast cancer diagnosis. Six categories of merging strengths were reported: walking together; praying together; seeking together information and supportive network; trusting together; adjusting together; being together. Spirituality was an integral component that influenced the effectiveness of each of these major categories.
Comez et al., 2016	TR	14	Qualitative	At least 1 year prior to recruitment	The process of women with BC and their spouses from diagnosis to treatment completion.	Different themes related to the phase of treatment—When the couples received the diagnosis, two main themes emerged: perceptions of breast cancer (BC) and reactions to BC. During the treatment process, four themes emerged: symptoms experienced, fear, understanding each other’s worth and needs and counseling. Three themes characterized the stage related to coping with the disease and treatment: process body image and sexuality, religious beliefs and support systems. After the treatment period, three themes were found: changes in roles, health-promoting behaviors and living for oneself and not for others.
Hoskins, 1995	USA	128	Observational	Six measurements points: at 7–10 days, atone, 2, 3, and 6 months, and 1 year post surgery	Adjustment outcomes as an interpersonal variable.	Emotional adjustment in both patients and partners could be predicted by satisfaction of interactional and emotional needs. The effects of cancer accentuated the dynamics of a complementary pattern of interaction (partners perceived and complemented i each other’s needs) as a strategy for coping with the experience. This interaction may be enhancing the likelihood of positive emotions.
Northouse et al., 1998	USA	73	Observational	Three measurements points: at the time of diagnosis and at 60 days and 1 year post diagnosis	The concurrent stress, resources, appraisal and patterns of adjustment of couples in the benign and malignant groups, comparing the psychosocial responses of patients versus spouses.	During the first year following diagnosis, the patterns of adjustment (emotional distress and role problems) were clearly dissimilar for couples facing benign versus malignant disease. Couples in the malignant group reported significantly higher levels of emotional distress and more role problems than couples in the benign group, and these differences persisted over time. Couples in the malignant group reported greater decreases in marital satisfaction and family functioning than couples in the benign group. A significant decline of social support over time was reported by women with malignant breast cancer. Couples in the malignant group reported more uncertainty about the nature and course of the illness than couples in the benign group. The uncertainty of women with breast cancer decreased over time, but their uncertainty and that one of their husbands remained markedly higher than the level of uncertainty reported by couples in the benign group.
Meier et al., 2019	OH	70	Observational	Three measurements points at 2 weeks, at 3 months, and 1 year after cancer surgery	The effect of Common Dyadic Coping (CDC) on individuals 1 year after cancer surgery. Psychological distress in patients and their partners.	At 1 year after cancer surgery, patients and partners reported lower psychological distress when the couple showed a CDC in terms of couples’ agreement on how partners cope as a couple. Specifically, high CDC congruence was related to lower psychological distress among female patients. However, CDC may lose its importance over time when couples cope with chronic issues related to the disease.
Manne et al., 2006	USA	127	Observational	Two measurement points: during cancer treatment and at 9 months after the baseline (Time 2)	The association between types of couple’s communication strategies and couple’s ability to handle breast cancer’ stressors.	Partners who reported more constructive mutual communication had lower levels of discomfort at Time 2, while partners who reported greater avoidance of discussing problems and stressors or greater use of request-withdrawal communication experienced higher levels of suffering. Greater communication regarding stressors due to illness between patient and partner reduced the partner’s anguish levels and increased relationship satisfaction while potentially modulating the perception of the patient’s physical damage. Greater mutual constructive communication was a significant predictor of low partner distress, and greater mutual omission was a marginally significant predictor of partner suffering. Furthermore, request-withdrawal communication was not a significant predictor of the partner’s distress.
Kim et al., 2008	USA	168	Observational	Approximately 2 years from the diagnosis prior to recruitment	The dyadic effects of psychological distress on the quality of life of couples dealing with cancer.	The strongest predictor of the couple’s quality of life was the individual psychological distress, and it was strongly related to mental health. At 2 years post diagnosis, cancer survivors and their spouses displayed normal levels of psychological wellbeing and quality of life. Patients and partners caregivers reported similar levels of psychological distress and quality of life. The partner’s distress and the (dis)similarity in the levels of distress of the couple played a significant role in the individual quality of life. At the dyadic level, for men, a greater dissimilarity in psychological distress was associated with better physical health.

## Data Availability

Not applicable.

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
