# Peer review of "Relationship Dynamics among Couples Dealing with Breast Cancer: A Systematic Review"

_ijerph, 2021, doi:10.3390/ijerph18147288_

Round 1

Reviewer 1 Report

Thank you for giving me the opportunity to review the manuscript entitled  Relationship dynamics among couples dealing with breast can-cer: a systematic review”

Please find below several comments which can improve the manuscript:

The topic of the manuscript is important and in the scope of the Journal however the manuscript requires significant improvement as in current form it is difficult to understand it.

The authors need to consider the careful checking of whole manuscript and improve it to be written in scientific form and using scientific terms/language (I.e. cancer journey looks very strange in scientific paper)

The abstract of the manuscript is limited and required abstention – with better methods, results and conclusions from this review

The sentence in line 23-26 is very general – more details should be given (please add some data)

It is written “Fig. 1 reports,…..” but I cannot  find such figure in the paper – please double check it

The authors can consider moving some tables into the supplementary file document

Author Response

Response to Reviewer 1 Comments

The authors need to consider the careful checking of whole manuscript and improve it to be written in scientific form and using scientific terms/language (i.e. cancer journey looks very strange in scientific paper).

Thank you for your very careful review of our paper and for the comments. They have been seriously considered and implemented as to improve the scientific quality of the abstract and the manuscript (see the author ,I.C, for the English implementations).

The abstract of the manuscript is limited and required abstention – with better methods, results and conclusions from this review

We agree with your comment that can help to better understand our study. Please see “Seventeen studies were included, and a synthesis of the literature revealed five domains that characterized the dyadic process: dyadic coping strategies, psychosocial support, communication, couple’s sexual life and spiritualit. The first domain concerned the coping strategies to face the problem. Couples who adopt common dyadic coping strategies report less psychological distress and high levels of adaptation to the pathology. The second dimension regarded the psychosocial support, both instrumental and emotional, aimed to assist the couple in maintaining a positive dyadic relationship, in the acute phase of the treatment and during the early survivorship phase. The third aspect focused on communication, i.e. a positive and complementary pattern of interactions which favor the dyad in facing the breast cancer care path and this, in turn, enhances the likelihood of positive emotions. The fourth theme concerned changes in the couple’s sexual life as consequence of the women’s personal transformations caused by treatments. The fifth theme illustrated the influence of spirituality on the marriage union. 

The sentence in line 23-26 is very general – more details should be given (please add some data)

Thank you for this comment. This aspect has now been explicitly addressed. Discuss pag 1 line 23-26: “In the past century, due to the increase in breast cancer incidence3 advances in the treatment of the disease have been achieved. Earlier diagnosis, effective screening programs and systemic anticancer therapies, therefore, lead to improved patients’ survival outcomes including disease-free survival and overall survival4.”

It is written “Fig. 1 reports,…..” but I cannot  find such figure in the paper – please double check it

Thank you for your comment. Probably, there was probably a problem uploading the file.

Reviewer 2 Report

This manuscript describes a systematic review of interest on breast cancer on the couple's relationship. However, the last systematic review was made in 2017, (following the references on the paper). I would like to know more about what is the need of doing this research, with a very low diference in years between this and the last one.

In the text, I suggest the following changes:

-L.23: changing "most women" for specific % or data.

-L.26-29: this paragraph reflects some ideas in only one sentence. The redaction should be revised to put it simpler and in more sentences.

-L.49: "20" should be made a reference.

-L.61-63: As stated before, the motives of the review should be a bit deeper.

-L.83-85: Exclusion criteria should be clarified or explained.

-L.91: Pubmed is an access to MEDLINE database, I suggest changing it by "MEDLINE (via Pubmed)"

Despite these changes, I found the study very interesting and with impact.

Author Response

This manuscript describes a systematic review of interest on breast cancer on the couple's relationship. However, the last systematic review was made in 2017, (following the references on the paper). I would like to know more about what the need is of doing this research, with a very low difference in years between this and the last one.

Thank you for this comment. The previus review focus the attetion on the factors (including demographic, disease-related and psychosocial factors) have been associated with marital adjustment in both patients and their partners. Moreover, the authors try to identify and describe which measures have been employed to assess marital adjustment in the context of BC. Conversely, we highlight the different relationship ‘areas/dynamics that the dyads enact during the breast cancer treatment path. Indeed, we aim to point out the ones that significantly impact the psychological adjustments to the BC. 

-L.23: changing "most women" for specific % or data.

-L.26-29: this paragraph reflects some ideas in only one sentence. The redaction should be revised to put it simpler and in more sentences.

The information has now been provided accordingly to your suggestion. In the past century, due to the increase in breast cancer incidence3 advances in the treatment of the disease have been achieved. Earlier diagnosis, effective screening programs and systemic anticancer therapies, therefore, lead to improved patients’ survival outcomes including disease-free survival and overall survival4. Traditionally, in the oncology literature, greater attention is paid to the psychiatric, psychological and psychosocial impact of the pathology. Several studies analyzed the adaptive reaction strategies, of patients, at the medically defined points such as diagnosis disclosure, treatment consents, disease recurrence and palliative cares initiation

-L.49: "20" should be made a reference.

We are sorry for the missing information. 20 it has been made as reference. Now the reference number 20 is included in the reference list.

-L.61-63: As stated before, the motives of the review should be a bit deeper.

Thank you for your comment. The motives of the review have now been provided. Research points14,16 out the detrimental effects of marital distress on patients. Based on this evidence, couples who experience the high-er levels of distress have poor conflict resolution skills, higher levels of dissatisfaction with the relationship, high degrees of conflict and hostility, and different perceptions and expectations about the disease among them. In this context, just few reviews have focused on the impact of breast cancer on the couple's relationship 15,18,24,25. They mainly reported the areas of interaction of breast cancer and the patient's personal relationships: 1) general interactional themes of patients sec-ondary to breast cancer (e.g. social isolation; victimization; uncertainty); 2) the impact of social support including the support from partners on patients’ psychological adjustment; 3) the impact of the disease on the parent-child relationship. Moreover, Staff 25 highlighted the dimension of the dyadic coping by exploring the antecedents and the outcomes in close personal relationships. In addition, these reviews discussed the impact of breast cancer on marital level of satisfaction in terms of couples’ sexual relations and commu-nication. Despite the emerging evidence on the impact of breast cancer on patients and partners, no systematic review has been conducted specifically on the relationship dy-namics that can have a major impact on the psychological distress experienced by couples

-L.83-85: Exclusion criteria should be clarified or explained.

Thank you for your comment. Integrations have been made in line with your suggestion. Articles were excluded if they focused on: (a) only one member of the dyad and/or patients (i.e., not population of interest); (b) dealt specifically with either support or intervention on couples (i.e., not context of interest); (c) causes of breast cancer, prevalence and incidence, medical considerations, evaluation and assessment of interventions (i.e., not outcome of interest); and (d) they did not report empirical findings. Due to the exploratory nature of this review, there were no restrictions on the type of data to look for and extract.

-L.91: Pubmed is an access to MEDLINE database, I suggest changing it by "MEDLINE (via Pubmed)"

The information has now been provided accordingly to your suggestion.

Reviewer 3 Report

I congratulate the authors for all the work. the theme is very interesting, knowing the dynamics of the dyad is of paramount importance.
However, improvement in methodology will be needed. I leave some tips that I'm sure will help. Use a good methodological framework on how to do a systematic review is very important. What is your methodological framework?

Abstract
The objective or question should be clear in the abstract.
the objective is to explore the couple's experience of breast cancer treatment?
When I read this sentence - "In this review, the couple experience of breast cancer treatment was investigated" 
I thought the objective might be that. However, when analyzing the results, I do not analyze this congruence. The results refer to strategies for coping/adjustment with the treatment process.
Align this kind of thinking. What kind of systematic review is it?
 Is it scoping? Is it the effectiveness? Is it the qualitative evidence?
Note the title -  have Relationship dynamic, in the abstract, they never use this concept, in the keywords too (or something similar). In the introduction, this concept was not defined.
Defined the methodology rigorously can help.
The keywords mentioned using psychological adjustment.  Psychological adjustment appears in introduction but is not defined. It is crucial in a review to have a clear definition of the central concepts.

Line 63 - the authors mentioned that the "aim of this review is to highlight and summarize the relationship dynamics that couples enact during the breast cancer treatment path, after the acute phase of treatments. Furthermore, we want to clarify which areas might predict a positive trajectory of psychological dyadic adjustment." 
Line 77 - But then the authors mentioned that used PICO.
The PICO mnemonic to construct a clear and meaningful review objective/question regarding the quantitative evidence on the effectiveness of interventions.
In the theoretical framework that I know – JBI - The PICO is for the effectiveness review, eg -What is the effect of inspiratory muscle training versus no specific training on dyspnea and functional ability? 
In this review what are your outcomes like an intervention. They are unclear. 
P - dyads - ok
I - what are the criteria for the intervention? and what will the intervention be?
C – comparison with what? 
O - ? 
PICO questions is for quantitative studies.
If it is PICo mnemonic stands for the Population, the Phenomena of Interest, and the Context - already a type of review that includes qualitative studies. However, scoping can involve qualitative and quantitative studies. 
Line 91 - Pubmed is not a database but a search engine.
Line 130 - Mention that figure 1 reports ... but I can't find the figure.
Line 146 - Table titles are swapped. Table two refers to the assessment of quantitative studies and table 3 to qualitative studies.
I no understand  table 4 of the article Mallen, C., Peat, G., & Croft, P. (2006). Quality assessment of observational studies is not commonplace in systematic reviews.  Journal of clinical epidemiology, 59(8), 765-769 , as Critical Appraisal tools. It is a compilation of parameters performed by the author.

References need to be revised, for example when referring to Prism 29 - the reference does not match. this is an example.

Suggestions for authors - define a clear objective (the simpler the better) and rethink the methodology of the systematic review.

Author Response

However, improvement in methodology will be needed. I leave some tips that I'm sure will help. Use a good methodological framework on how to do a systematic review is very important. What is your methodological framework?

Line 63 - the authors mentioned that the "aim of this review is to highlight and summarize the relationship dynamics that couples enact during the breast cancer treatment path, after the acute phase of treatments. Furthermore, we want to clarify which areas might predict a positive trajectory of psychological dyadic adjustment." 
Line 77 - But then the authors mentioned that used PICO.

Thank you for bringing this issue to our attention. The reference of PICO was a mistake. A notes left-over inadvertently.

We performed a narrative synthesis of the literature in this filed. This point has now been addressed, highlining the methodological nature of our review.

Line 130 - Mention that figure 1 reports ... but I can't find the figure.

Thank you for your comment. Probably, there was probably a problem uploading the file, not the figures is displayed.

Line 146 - Table titles are swapped. Table two refers to the assessment of quantitative studies and table 3 to qualitative studies.

We are sorry for that we amended our mistake and sorted the table tiles correctly.

Round 2

Reviewer 1 Report

The manuscript improved, however I still cannot see the figure. What is more the authors ignore me comment "The authors can consider moving some tables into the supplementary file document".

I am still convinced that some of the tables can be included as SM.

Author Response

Thank you for your comment. We are very sorry to realize that there was an error loading the manuscript with the figure mentioned in the text. We will upload it again and we hope it will be okay.

Your suggestion to include some tables in the supplementary materials is perfectly adequate. It was certainly not our intention to ignore your comment. However, since  these tables show the methodological and qualitative steps of our systematic review, we believe it is necessary to keep them into the body of the article, in order to highlight the quality of the work done.
